

# Validation of carbon isotope fractionation in algal lipids as a $PCO_2$ proxy using a natural $CO_2$ seep (Shikine Island, Japan)

Caitlyn R Witkowski[1], Sylvain Agostini[2], Ben P Harvey[2], Marcel TJ van der Meer[1], Jaap S Sinninghe Damsté[1,3], Stefan Schouten[1,3]

[1]Department of Marine Microbiology and Biogeochemistry, Royal Netherlands Institute for Sea Research, Den Burg (Texel), 1790AB, The Netherlands
[2]Shimoda Marine Research Center, University of Tsukuba, Shimoda, 415-0025, Japan
3Department of Geosciences, Utrecht University, Utrecht, 3508 TA, The Netherlands

*Correspondence to*: Caitlyn R Witkowski (caitlyn.witkowski@nioz.nl)

**Abstract.** Carbon dioxide concentrations in the atmosphere play an integral role in many earth system dynamics, including its influence on global temperature. Long-term trends can provide insights into these dynamics though reconstructing long-term trends of atmospheric carbon dioxide (expressed in partial pressure; $PCO_2$) remains a challenge in paleoclimatology. One promising approach for reconstructing past $PCO_2$ utilizes isotopic fractionation associated with $CO_2$-fixation during photosynthesis into organic matter ($\varepsilon_p$). Previous studies have focused primarily on testing estimates of $\varepsilon_p$ derived from species-specific alkenone compounds in laboratory cultures and mesocosm experiments. Here, we analyze $\varepsilon_p$ derived from general algal compounds from sites at a $CO_2$ seep near Shikine Island (Japan), a natural environment with $CO_2$ concentrations ranging from ambient (ca. 310 $\mu$atm) to elevated (ca. 770 $\mu$atm). We observed strong, consistent $\delta^{13}C$ shifts in several algal biomarkers from a variety of sample matrices over the steep $CO_2$ gradient. Of the three general algal biomarkers explored here, namely loliolide, phytol, and cholesterol, $\varepsilon_p$ positively correlates with $PCO_2$ in agreement with $\varepsilon_p$ theory and previous culture studies. $PCO_2$ reconstructed from the $\varepsilon_p$ of general algal biomarkers show the same trends throughout, as well as the correct control values, but with lower absolute reconstructed values than the measured values at the elevated $PCO_2$ sites. Our results show that naturally-occurring $CO_2$ seeps may provide useful testing grounds for $PCO_2$ proxies and that general algal biomarkers show promise for reconstructing past $PCO_2$.

## 1 Introduction

The current increase in the atmospheric concentration of carbon dioxide (expressed in partial pressure, $PCO_2$) plays a leading role in climate change (Forster et al., 2007). $PCO_2$ is significantly higher now than it has been in the past 800 ka (Lüthi et al., 2008) and although long-term changes in $PCO_2$ are not uncommon over millions of years (Foster et al., 2017), this current spike in $PCO_2$ has occurred within only the past two centuries (IPCC, 2013). Uncertainties remain on the exact magnitude to which $PCO_2$ influences climate, as well as the exact response of the environment to these climate changes. Long-term $PCO_2$ trends help us better understand the context for these changes and are reconstructed via indirect means, i.e. environmental



proxies. However, current proxies leave much to be desired, often with large uncertainties and conflicting values over the geologic record (Foster et al., 2017). The development of new proxies for $PCO_2$ may help us better constrain past trends and provide the critical data needed to build robust models of future climate.

One proxy that has been explored with mixed success over the past several decades is the stable carbon isotopic fractionation
associated with photosynthetic inorganic carbon fixations ($\varepsilon_p$), which has been shown to positively correlate with $PCO_2$ (Bidigare et al., 1997; Jasper and Hayes, 1990; Zhang et al., 2013). $\varepsilon_p$ occurs as the $CO_2$-fixing enzyme in photoautotrophs, Rubisco (ribulose 1,5-biphosphate carboxylase oxygenase), favors $^{12}C$ which consequently results in photosynthates isotopically lighter than the original carbon source. A greater abundance of $CO_2$ increases Rubisco-based isotopic discrimination, resulting in an even lower $^{13}C/^{12}C$ ratio ($\delta^{13}C$) in photoautotroph biomass (Farquhar et al., 1989; Farquhar et
al., 1982; Francois et al., 1993; Popp et al., 1989). When this phototrophic biomass is preserved in the geologic record, the $\delta^{13}C$ of sedimentary organic matter can be used to reconstruct $PCO_2$ (Hayes et al., 1999). The largely mixed contributions and diagenetic processes on bulk organic matter can, however, mask this signal (Hayes, 1993; Hayes et al., 1999). Thus, isotope analysis of specific biomarker lipids is preferred in order to better define the source of the $\delta^{13}C$ signal (Jasper and Hayes, 1990; Pagani, 2002).

The most studied biomarkers for calculating $\varepsilon_p$ are alkenones, i.e. long-chain unsaturated methyl and ethyl ketones produced by select Haptophytes (de Leeuw et al., 1980; Volkman et al., 1980). The stable carbon isotopic fractionation of alkenones has been studied using cultures and mesocosms with controlled environments (Benthien et al., 2007; Laws et al., 1995), but conditions do not always mimic natural environments and the natural variation in carbonate chemistry that occurs on a daily to seasonal time scales. Furthermore, these experiments are also time-consuming given that they must have delicately
balanced water chemistry including $CO_{2[aq]}$ concentrations, pH, and alkalinity, as well as nutrients such as nitrate and phosphate (Bidigare et al., 1997; Laws et al., 1995; Popp et al., 1998), along with the additional challenge of maintaining a constant $\delta^{13}C$ of the $CO_{2[aq]}$ while photoautotrophs continually enrich the growth water as they fix $CO_2$. Water column studies (Bidigare et al., 1997) and surface sediments (Pagani, 2002) have been applied but rarely reach elevated $PCO_2$ levels such as those encountered in the past.

Here we use an alternative approach by analyzing algal lipids near natural $CO_2$ seep systems. In tectonically active zones, volcanically-induced seeps consistently bubble high $PCO_2$ concentrations into the surrounding water, substantially increasing the local $CO_2$ concentrations in the water and providing an environment referent to past and future high-$CO_2$ worlds. $CO_2$ seeps were previously overlooked for biological studies due to the typically high sulfide ($H_2S$) concentrations associated with volcanic degassing that make these environments largely uninhabitable (Dando et al., 1999). However,
certain $CO_2$ seep systems have been found to have low $H_2S$ making them suitable for ocean acidification experiments (Hall-Spencer et al., 2008), prompting further research in e.g. the Mediterranean (Boatta et al., 2013), in Japan (Agostini et al., 2015), Papua-New-Guinea (Fabricius et al., 2011), and New Zealand (Brinkman and Smith, 2015). These sites may provide an ideal testing ground for the impact of isotopic fractionation on algal lipids where environmental conditions are at naturally balanced levels with the exception of the large gradient of $CO_2$ concentrations.





In our study, we explore the relationship between $\mathcal{E}_p$ and $CO_{2[aq]}$ across several pre-established sites with different (temporally consistent) levels of $PCO_2$ at the warm-temperate $CO_2$ seep at Mikama Bay off the shore of Shikine Island, Japan. We test this relationship using general algal biomarkers, compounds which have rarely been used for $\mathcal{E}_p$-based $PCO_2$ reconstructions despite their potential utility (Freeman and Hayes, 1992; Popp et al., 1989; Witkowski et al., 2018).

**2 Materials and Methods**

**2.1 Sample site**

The site is briefly described here. For further details, we refer to Agostini et al. (2018). Mikama Bay is located on the south-southwest corner of Shikine Island off the Izu Peninsula, Japan with several venting locations in the north of the bay (34.320865 N, 139.204868 E). The gas emitted from the seep contains 98% $CO_2$ and the bay has a spatially and temporally

constant total alkalinity averaging at 2265 ± 10 $\mu$mol kg$^{-1}$. Samples were collected from three preestablished $PCO_2$ sites (Agostini et al. 2018; Harvey et al. 2018), "Control $PCO_2$" site in an adjacent bay outside the influence of the $CO_2$ seep ($PCO_2$ 309 ± 46μatm), a "Mid $PCO_2$" site ($PCO_2$ ca. 460 ± 40 $\mu$atm), and a "High $PCO_2$" site ($PCO_2$ 769 ± 225) ( Agostini et al., 2018; Harvey et al., 2018; Fig. 1). Temperature (annual range ca. 14 to 28ºC) and salinity (ca. 34‰) are relatively homogenous throughout the bay and do not differ between the elevated $PCO_2$ sites and control $PCO_2$ sites (Agostini et al.

2018). Currents and wind were measured in October 2014 and April 2015 (Agostini et al., 2015). October 2014 measurements showed moderate turbulent winds (ranging from 0.6-11.5 m s$^{-1}$, average 4.5 m s$^{-1}$) associated with current velocities (ranging from 0 to 1.6 knots, average 0.4 knots) at 5 m in the surface water, whereas April 2015 measurements showed moderate north-northeast winds (1.5-8.6 m s$^{-1}$, average 5.1 m s$^{-1}$) associated with low current velocities (0-0.2 knots, average 0.04 knots).

**2.2 Materials**

Samples were collected in June and September of 2016 (indicated in Fig. 1). All samples were collected in at least triplicate for each site ("Control $PCO_2$", "Mid $PCO_2$" and "High $PCO_2$" site). Additional control sites (at ca. 1.8 km and 2.4 km away from the $CO_2$ seep) around the island were taken to ensure the fidelity of the Control site closest to the seep. June sampling included surface waters for dissolved inorganic carbon (DIC) measurements, surface sediments, and benthic diatoms

attached to surface sediment through extracellular polymeric substance production. In September, macroalgae, SPM, and plankton tows were collected, in addition to surfaced water DIC and surface sediments, taken in triplicate at each site on four separate days.

Water for the δ$^{13}$C of DIC analysis was collected by overfilling glass vials with sea surface water and adding mercury chloride (0.5%) before closing with a septa cap and sealing with electrical tape. Surface sediments were collected by divers

using geochemical sample bags. Macroalgae and benthic diatoms were scraped off submerged rocks at each respective site. SPM was sampled by collecting sea surface water in three 23 L Nalgene tanks (20 L filtered from each tank) and taken back



to the lab where 60 L per site per day were filtered over a single 0.7 $\mu$m GFF (combusted prior to sampling for 3 h at 300°C). A 25 $\mu$m mesh plankton net ("small plankton net", Rigo, Saitama, Japan) was towed 50 m three times per site and filtered using a portable hand aspirator on the boat over a single 47 mm muffled GFF. All samples were immediately frozen; once back in the lab, these were freeze-dried and kept in a refrigerator.

## 5 2.3 Methods

Each seawater sample was measured for the $\delta^{13}C$ of DIC in duplicate on a Thermo Scientific Gas Bench II coupled to a Thermo Scientific Delta V mass spectrometer. Prior to running samples, 12 ml vials were prepared with 100 $\mu$L of 85% $H_3PO_4$ and flushed with He. 500 $\mu$L of seawater was then added and left to react for at least 1 h prior to measuring the headspace. Standards were run at the start, end, and every six runs of a sequence. Standards were prepared with 0.3 mg of

$Na_2CO_3$ and 0.4 mg of $Ca_2CO_3$ (all calibrated against NBS-19) which were then flushed with He, injected with 500 $\mu$L of 85% $H_3PO_4$, and reacted for 1 h. The headspace was then measured. Average values and standard deviation errors reported are based on six measurements for June (three at the High $P$CO$_2$ site and three at the Control) and thirty-six measurements for September (three each at the High PCO$_2$ site, Mid PCO$_2$ site, and Control collected on four separate days).

Freeze-dried sediments, benthic diatoms, and macroalgae were homogenized using mortar and pestle and extracted using a

Dionex 250 accelerated solvent extractor at 100°C, 7.6x106 Pa using dichloromethane (DCM): MeOH (9:1 v/v). GFFs containing plankton net material or SPM were cut into 1 mm x 1 mm squares and extracted using ultrasonication with 2 ml dichloromethane (DCM): MeOH (9:1 v/v) five times. All total lipid extracts (TLEs) were hydrolyzed by refluxing the TLE with 1N of KOH in MeOH for one hour and neutralized to pH 5 using 2 N of HCl in MeOH. Bi-distilled water (2 ml) and DCM (2 ml) were added (5x) to the hydrolyzed centrifuge tubes and the DCM layers with pooled organic matter was

removed and dried over $Na_2SO_4$. After drying the resulting base hydrolyzed TLE, samples were eluted over an alumina packed column into apolar (hexane: DCM, 9:1 v/v), ketone (DCM), and polar (DCM: MeOH, 1:1 v/v) fractions. Polar fractions were then silylated using pyridine: N,O-Bis(trimethylsilyl)trifluoroacetamide (BSTFA) (1:1 v/v) and heated at 60°C for 1 h prior to running on the gas chromatography-flame ionization detector (GC-FID), gas chromatography-mass spectrometry (GC-MS), and gas chromatography isotope-ratio mass spectrometry (GC-IRMS).

Silylated polar fractions were analyzed by GC-FID to check the baseline quality of the sample and to determine quantities of compounds. Based on the quantities, fractions were diluted with ethyl acetate to prepare ideal concentrations for GC-MS to identify compounds and for GC-IRMS to measure the isotopic composition of specific compounds. Each instrument is equipped with the same CP-Sil 5 column (25 m x 0.32 mm; df 0.12 μm) and He carrier gas. GC-FID, GC-MS, and GC-IRMS had starting oven temperatures of 70ºC ramped at 20ºC/min to 130ºC and then ramped at 4ºC/min to 320ºC for 10

min. All three instruments use the same in-house mixture of $n$-alkanes and fatty acids to check chromatography performance at the start of each day (GC-standard). For compound specific stable carbon isotope analysis using GC-IRMS, additional standards with known isotopic values (-32.7 and -27.0‰) of per deuterated (99.1%) $n$-alkanes (C$_{20}$ and C$_{24}$, respectively), were co-injected with the GC-standard. Samples were also co-injected with the C$_{20}$ and C$_{24}$ GC-IRMS standards to monitor





instrument performance. Every day, the Isolink II combustion reactor of the GC-IRMS was oxidized for at least 10 min, backflushed with He for 10 min, and purged for 5 min; a shorter version of this sequence is conducted in post-sample seed oxidation which includes 2 min oxidation, 2 min He backflush, and 2 min purge conditioning line. Longer oxidations were run weekly. Each derivatized compound was corrected for the $\delta^{13}C$ of the BSTFA used in silylation (-32.2‰).

**3 Results**

Samples from different matrices were collected at several Control $PCO_2$ sites (309 ± 46, at a "Mid $PCO_2$" site (ca. 100 m from the venting area; 460 ± 40 $\mu$atm), and near the venting area ("High $PCO_2$" site; 769 ± 225 $\mu$atm) during June and September 2016 (Fig. 1), which included June-collected surface waters (for DIC), surface sediments, and benthic diatoms, and September-collected surface waters (for DIC and SPM), surface sediments, plankton net tows, and macroalgae. With the

exception of the SPM from surface waters, all samples yielded enough material for isotope studies, and therefore phytoplankton filters from surface waters were not included in this study.

The $\delta^{13}C$ of DIC demonstrated minimal change over the gradient of $CO_2$ and minimal change between the two seasons (Fig. S1). The June $\delta^{13}C$ of DIC was 0.2 ± 0.2 ‰ (± SD; N=3) at the Control site and 0.5 ± 0.04 ‰ (N=3) at the High $PCO_2$ site. The September $\delta^{13}C$ of DIC was -0.4 ± 0.2 ‰ (N=8) at the Control site, -0.1 ± 0.1 ‰ (N=8) at the Mid $PCO_2$ site, and 0.2 ±

0.4 ‰ (N=8) at the High $PCO_2$ site.

The polar fractions of the extracts of the surface sediments, plankton, macroalgae, and benthic diatoms showed a similar suite of compounds, observed across all sites and during both seasons. The most prominent compounds were loliolide, phytol, $C_{14}$-$C_{16}$ alkanols, and sterols such as cholesta-5,22E-dien-3β-ol, cholesterol, 23-methylcholesta-5,22dienol, campesterol, stigmasterol, and β-sitosterol (e.g. Fig. 2). Loliolide, phytol, and cholesterol were targeted for isotope analysis

as the most abundant general algal biomarkers and with relatively good separation in the GC.

Among the sample matrices, the $\delta^{13}C$ of loliolide ranges from -19.8 to -22.0 ‰ at the Control sites, from -20.5 to -22.9 ‰ at the Mid $PCO_2$ site, and from -23.1 to -29.0 ‰ at the High $PCO_2$ site (Fig. 3A). The $\delta^{13}C$ of loliolide from June surface sediments shows the strongest isotopic change from the Control site to the High $PCO_2$ site (-21.2 to -29.0 ‰), followed by the $\delta^{13}C$ of loliolide from September macroalgae (-21.3 to -25.7 ‰). A lesser $\delta^{13}C$ shift is observed in the September surface

sediment-derived loliolide (-19.8 to -23.1 ‰). The $\delta^{13}C$ of the benthic diatom-derived loliolide (-20.2 to 23.6 ‰) and the plankton tow-derived loliolide show the smallest isotopic shifts from the Control to High $PCO_2$ site (-22.0 to -23.6 ‰).

Similar to the results of the $\delta^{13}C$ of loliolide, the $\delta^{13}C$ of phytol also consistently shows higher $\delta^{13}C$ values in the Control sites and lower $\delta^{13}C$ values in the elevated $PCO_2$ sites among all samples types collected in both seasons (Fig. 3B). For the whole sample set, the $\delta^{13}C$ of phytol ranges from -18.9 to -22.6 ‰ at the Control site, from -19.4 to -22.4 ‰ at the Mid

$PCO_2$ site, and from -22.6 to -27.8 ‰ at the High $PCO_2$ site (Fig 3B), similar ranges as observed for loliolide. A similar shift in $\delta^{13}C$ values of phytol is observed with increasing $PCO_2$ in the June surface sediments (-22.6 to -27.8 ‰), the June benthic



diatoms (-18.9 to -24.4 ‰), and the September macroalgae (-21.5 to -26.9 ‰). Smaller differences in the $\delta^{13}C$ of phytol are observed for September plankton (-21.7 to -24.4 ‰) and September sediment (-20.5 to -22.6 ‰).

The $\delta^{13}C$ of cholesterol likewise shows a similar trend to the other two biomarkers but with a smaller shift in the $\delta^{13}C$ values from the Control $PCO_2$ sites to the elevated $PCO_2$ sites. Among the different sample matrices, the $\delta^{13}C$ of cholesterol ranges

from -21.2 ‰ to -25.1 ‰ at the Control site, -22.1 to -23.4 ‰ at the Mid $PCO_2$ site and -23.1 to -27.4 ‰ at the High $PCO_2$ site (Fig. 3C). The strongest change in the $\delta^{13}C$ of cholesterol with increase $PCO_2$ occurs in the June surface sediments from -22.6 ‰ in the Control to -27.8 ‰ at the High $PCO_2$ site. The June benthic diatoms also have a large isotopic shift in the $\delta^{13}C$ of cholesterol (-21.2 to -25.8 ‰), as does the September macroalgae (-22.7 to -25.8 ‰). The September surface sediments (-22.2 to -23.1 ‰) and plankton tow-derived cholesterol (-25.1 to -26.7 ‰), however, have a smaller shift from

the control to the elevated $PCO_2$ sites.

## 4 Discussion

### 4.1 The $\delta^{13}C$ differences in biomarkers among matrices and seasons

All three biomarkers, phytol, loliolide and cholesterol, show a negative shift in $\delta^{13}C$ values with increasing $PCO_2$ in each matrix and each season (Fig. 3), agreeing with the theory that higher $PCO_2$ conditions result in lighter $\delta^{13}C$ values in biomass

(Farquhar et al., 1982). However, despite all having algal sources, the absolute isotope values vary for each compound, matrix, and season.

First, absolute values vary among the three compounds. This may be expected given the different biosynthetic pathways leading to formation of each compound, as well as the different contributors to each compound. Loliolide, considered a diatom biomarker in paleoreconstructions (e.g. Castaneda et al., 2009), is the diagenetic product of fucoxanthin (Klok et al.,

1984; Repeta, 1989), a xanthophyll which contributes to approximately 10% of all carotenoids found in nature (Liaaen-Jensen, 1978). Phytol, considered a photoautotroph biomarker in paleoreconstructions (e.g. Hayes et al., 1990), is the side-chain of the vital and omnipresent pigment chlorophyll $a$ that directly transfers sunlight energy into the photosynthetic pathway in nearly all photosynthetic organisms. Sterols, considered a general eukaryotic biomarker in paleoreconstructions, are the eukaryotic tetracyclic triterpenoid lipids used for critical regulatory roles of cellular functions e.g. maintaining

membrane fluidity (Nes et al., 1993). Although sterols are virtually restricted to eukaryotes, some exceptions have been found in bacteria (e.g. Wei et al., 2016). Here we only examine cholesterol which is universally absent in prokaryotes and composes of up to 20-40% of eukaryotic plasma membranes (Mouritsen and Zuckermann, 2004). Phytol and cholesterol may have terrestrial sources given that they are derived from all photoautotrophs and all eukaryotes, respectively. However, the close resemblance of the isotopic composition among all three compounds, including the primarily diatom-limited

compound loliolide, suggests that these compounds share relatively similar source organisms. Furthermore, chlorophyll is rapidly degraded and does not readily transport, so although this site is coastal, it is unlikely that phytol is terrestrial-derived.



Cholesterol shows a lessened isotopic shift than the other two compounds from the ambient to elevated $PCO_2$ sites. Although this could also be due to terrestrial input, it more likely due to the mobile eukaryotic zooplankton in the water column which also contribute to the cholesterol signal.

Within the same biomarker and same season, some differences among matrices were observed. This difference may be due to the mobility of the matrix, as well as the algal assemblages. The plankton tow which captured free-floating surface water algae from that specific growth season is more readily transported by wind than the surface sediment, which likely reflects the culmination of multiple growth seasons throughout the water column. This is seen, for example, in the $\delta^{13}C$ of cholesterol collected in September from the same Control site where surface sediments are -22.2 ‰ and plankton tows are -25.1 ‰, where the latter has possibly been transported from sites with elevated $PCO_2$ levels. Similar differences among matrices are also observed in phytol and loliolide. The hypothesis of transportation affecting the isotopic signal in certain matrices is supported by the results from the macroalgae. The macroalgae, in contrast to the algae collected by plankton tows, were unaffected by transportation due to being fixed to the nearby rocks at each site. Thus, the isotopic composition of compounds of the macroalgae was similar to that of the long-accumulated surface sediments, e.g. -22.7 ‰ for the $\delta^{13}C$ of cholesterol at the September Control site.

Finally, there is a difference in the $\delta^{13}C$ values for biomarkers between seasons. The June-collected surface sediments and algae yielded a larger difference in $\delta^{13}C$ values along the $CO_2$ gradient than the September-collected surface sediments and algae. This seasonal difference may be due to extreme weather conditions experienced between the two sampling campaigns. Although typhoons are common in this region, in the weeks preceding the fieldwork in September, Shikine Island experienced an unusually high quantity of storms. The storms were also of unusual strength for this region of the Pacific, including Typhoons Mindulle and Kompasu, the severe tropic storms Omais and Chanthu, and the long-lived, erratic Lionrock typhoon. This atypical abundance and severity of storms observably ripped corals out of the rocks around Shikine Island and thus likely resuspended and transported some sediment around the bay. This would explain the reduced $\delta^{13}C$ difference between the Control and High $PCO_2$ site in the surface sediments collected in September, as well as the readily transportable algae collected by the plankton tow, and would explain why the rock-affixed macroalgae, also collected in September, maintained a strong $\delta^{13}C$ change across the transect.

### 4.2 The $\varepsilon_p$ among general algal biomarkers

To further validate the impact of $PCO_2$, we calculated the isotopic fractionation of algal biomass based on the $\delta^{13}C$ of the three biomarkers. Here we focus on surface sediments as they are a close analogue to the geological sediment records. Although the macroalgae and benthic diatoms also show strong isotopic fractionation, they represent a limited number of species and a single growth season. Furthermore, we calculated the $\varepsilon_p$ from the June-collected surface sediments, which appear to be the least affected by typhoon activity and represent fractionation over multiple seasons.

To calculate $\varepsilon_p$ in the June-collected surface sediments, we correct the $\delta^{13}C$ of the organic matter ($\delta_p$) for the $\delta^{13}C$ of the inorganic carbon source for the producers of these compounds ($\delta_d$) in Eq. (1):





$$\varepsilon_p = 1000 \cdot [ (\delta_d+1000) / (\delta_p+1000) - 1 ] , \hspace{3cm} (1)$$

$\delta_p$ is calculated by correcting the $\delta^{13}C$ for each individual biomarker for the offset with photosynthetic biomass caused by isotopic fractionation during biosynthesis. The isotopic offset between phytol and biomass is $3.5 \pm 1.3$ ‰ based on the average of twenty-three species compiled in (Witkowski et al., 2018) and the isotopic offset between sterols and biomass is

$4.5 \pm 3.0$ ‰ based on the average of eight algal species (Schouten et al., 1998). The isotopic offset for loliolide from biomass, however, has not been studied. Because isoprenoids are formed from the same biosynthetic pathway, we here average the offset of the other two isoprenoids here (4.0 ‰) to estimate a value for the difference between loliolide and biomass.

$\delta_d$ is calculated by correcting the measured $\delta^{13}C$ of DIC for temperature (Mook, 1974) and pH (Madigan et al., 1989), which

considers the relative contribution of different inorganic carbon species to the measured DIC. Based on the equations of Mook et al. (1974), we correct for the temperature-dependent carbon isotopic fractionation of dissolved $CO_2$ with respect to $HCO_3^-$ using the annual mean sea surface temperature for Shikine Island of 20.4ºC (Agostini et al., 2018). Based on the equations of Madigan et al. (1989), we corrected for the $\delta^{13}C$ of $HCO_3^-$ and $\delta^{13}C$ of $CO_{2[aq]}$ mass balance calculation that accounts for the relative abundance of these inorganic carbon species based on pH (Lewis and Wallace, 1998) at the High

$PCO_2$ site (7.81 $pH_T$) and Mid $PCO_2$ site (7.99 $pH_T$) relative to the ambient Control (8.14 $pH_T$). The corrected $\delta_d$ values yield -10.1 ‰ at the Control site, -10.0 ‰ at the Mid $PCO_2$ site, and -9.5 ‰ at the High $PCO_2$ site.

$\varepsilon_p$ values consistently yield much higher values at the elevated $PCO_2$ sites than the ambient Control sites in all three biomarkers, which share similar trends and absolute values (Fig. 4). $\varepsilon_p$ derived from loliolide averages $7.2 \pm 1.6$ ‰ at the Control, $9.2 \pm 1.6$ ‰ at the Mid $PCO_2$ site, and $15.9 \pm 1.6$ ‰ at the High $PCO_2$ site, $\varepsilon_p$ derived from phytol at $8.6 \pm 0.4$ ‰,

$8.6 \pm 0.9$ ‰, and $14.9 \pm 1.0$ ‰, respectively, and $\varepsilon_p$ derived from cholesterol at $7.6 \pm 3.0$ ‰, $9.2 \pm 3.1$ ‰, to $13.7 \pm 3.1$ ‰, respectively. Given that maximum fractionation for algae species is ca. 25 to 28 ‰ in laboratory cultures (Goericke and Fry, 1994), the $CO_2$ seep values suggests strong, but not close to maximum, fractionation of the local algae. These results show that $CO_2$ has a profound impact on $\varepsilon_p$ as the only variable with a large gradient in the bay.

### 4.3 $PCO_2$ reconstructed from general algal biomarkers

We estimate $PCO_2$ from the $\varepsilon_p$ values, a relationship first derived for higher plants (Farquhar et al., 1989; Farquhar et al., 1982) and later adapted for algae (Jasper and Hayes, 1990; Popp et al., 1989) in Eq. (2):

$$PCO_2 = [ b / (\varepsilon_f - \varepsilon_p) ] / K_0 , \hspace{3cm} (2)$$

where $\varepsilon_f$ reflects the maximum Rubisco-based isotopic fractionation, $b$ reflects species carbon demand per supply such as growth rate and cell-size (Jasper et al., 1994), and $K_0$ reflects a constant to convert $CO_{2[aq]}$ to $PCO_2$ based on temperature and

salinity (Weiss, 1974). $\varepsilon_f$ for algal species range from 25 to 28 ‰ in laboratory cultures (Goericke and Fry, 1994); we use an average 26.5 ‰ with an uncertainty of 1.5 ‰ uniformly distributed for these general algal biomarkers (Witkowski et al., 2018). The $b$ value is difficult to estimate as it is a catchall for factors other than $PCO_2$ that affect fractionation and is particularly difficult to estimate for general algal biomarkers because they are derived from a multitude of species. Previous





studies using phytol's diagenetic product phytane as a $PCO_2$ proxy (Bice et al., 2006; Sinninghe Damsté et al., 2008; van Bentum et al., 2012) have used a mean value of 170 ‰ kg $\mu M^{-1}$, similar to the mean of alkenone-producers. This is supported by a compilation of the $\delta^{13}C$ values of modern surface sediment organic matter mean average of 168 ± 43 ‰ kg $\mu M^{-1}$ (Witkowski et al., 2018) and a single study on phytol in the equatorial Pacific Ocean (Bidigare et al., 1997). We apply

this average, rounded to 170 ± 50 ‰ kg $\mu M^{-1}$ to all three general algal biomarkers.

The resulting reconstructed $PCO_2$ estimations show the expected values in the Control sites and much higher values in the elevated $CO_2$ sites among all three biomarkers (Fig. 5). Loliolide shows the biggest shift, from 239 +50/-49 $\mu$atm at the Control, 266 +57/-54 $\mu$atm at Mid $PCO_2$ site, and 437 +113/96 $\mu$atm at the High $PCO_2$ site. Phytol has similar but a slightly smaller shift in $PCO_2$ estimates to loliolide, with estimations of 264 +55/54 $\mu$atm, 291 +56/53 $\mu$atm, and 444 +98/87 $\mu$atm at

the Control, Mid $PCO_2$ site, and High $PCO_2$ site, respectively. Cholesterol shifts similarly to the other two biomarkers with 244 +67/-54 $\mu$atm, 266 +77/61 $\mu$atm, and 358 +136/90 $\mu$atm, respectively. These reconstructed values closely match each other and trend in the same direction as the actual values.

The reconstructed $PCO_2$ values derived from the $\delta^{13}C$ of general algal biomarkers closely match the actual measured $PCO_2$ values of the Control (Fig. 5), i.e. 309 ± 46 $\mu$atm (Agostini et al., 2018; Harvey et al., 2018), when considering the

uncertainty in the reconstructed estimations. However, the proxies underestimate the absolute values measured at the elevated $PCO_2$ sites (Fig. 5), i.e. 460 ± 40 $\mu$atm at the Mid $PCO_2$ site and 769 ± 225 $\mu$atm at the High $PCO_2$ site (Agostini et al., 2018; Harvey et al., 2018). This underestimation may be caused by some site limitations, including allochtonous input of sediment at the Mid and High $PCO_2$ site. This input seems likely given the intense weather conditions that occur annual in this small bay in which lateral transport of sediment could bring algal material grown in ambient $PCO_2$ conditions into the

bay and dampen the overall $PCO_2$ signal picked up in the biomarkers. Future research conducted at another $CO_2$ seep settings with different weather and current conditions could illuminate this.

**5 Conclusion**

We analyzed the $\delta^{13}C$ of general algal biomarkers in surface sediments, plankton, benthic diatoms, and macroalgae collected in a transect from a $CO_2$ vent during two seasons. The strong $\delta^{13}C$ change between the Control and elevated $PCO_2$ sites

suggest that the increased $CO_2$ concentrations in the seawater does indeed influence fractionation of photoautotrophic biomass and validates previous $PCO_2$ reconstructions which have considered utilizing general algal biomarkers for this purpose. Reconstructions correctly estimate control values, though reconstructions at the elevated $PCO_2$ sites show underestimations of the actual $PCO_2$, likely due to the allochtonous input. Our results show that $CO_2$ seeps may offer testing grounds for exploring new $PCO_2$ proxies under natural conditions at high $PCO_2$ levels such as those encountered in the

geological past.





**Acknowledgements**

We thank Jason M. Hall-Spencer, Marco Milazzo and Yasutaka Tsuchiya for their help in sample-collection. We also thank Jort Ossebaar and Ronald van Bommel at the NIOZ for technical support. **Funding**: This study received funding from the Netherlands Earth System Science Center (NESSC) through a gravitation grant (024.002.001) to JS and SS from the Dutch

Ministry for Education, Culture and Science. **Author contributions**: CRW, SS, and JSSD designed the study. SA, BPH, and CRW collected field samples. CRW analyzed samples and wrote the first draft of the manuscript. CRW, MvdM, and SS interpreted the data. **Competing interests**: The authors declare that they have no competing interests. **Data and materials availability**: All data are present in the paper and/or the Supplementary Materials.

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



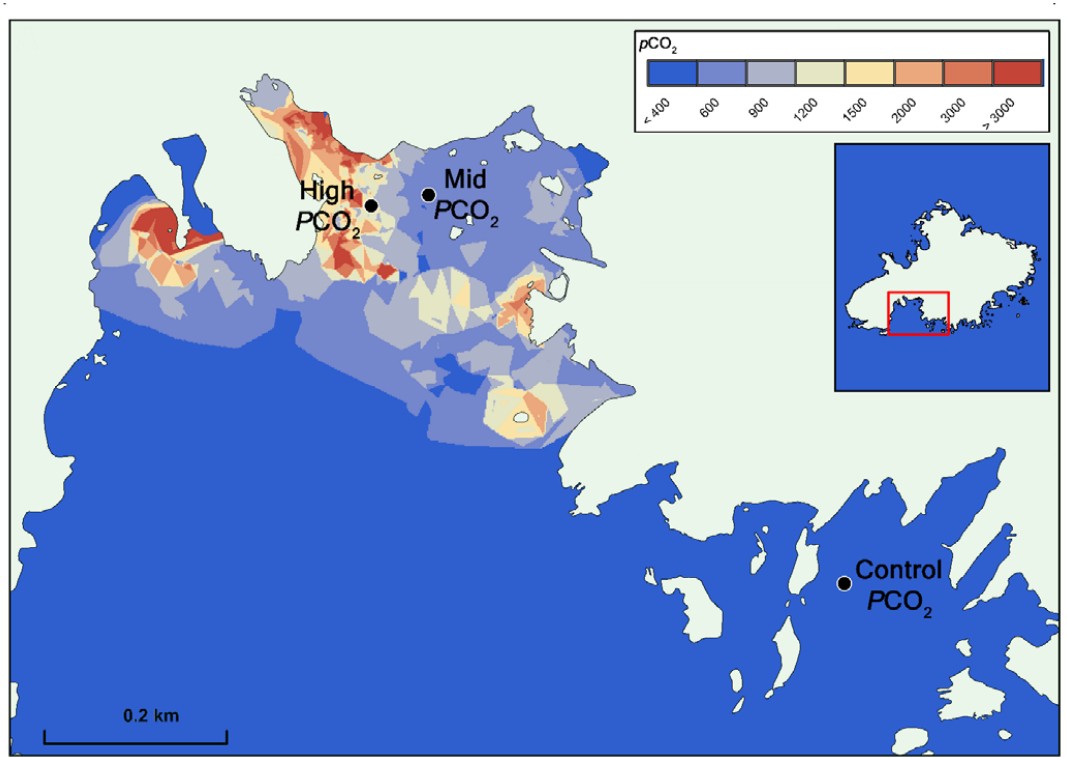

**Figure 1: Map of $PCO_2$ in the study region at Shikine Island (Japan).** Spatial variability in $PCO_2$ (Agostini et al., 2018), computed using the nearest neighbor algorithm in ArcGIS 10.2 software (http://www.esri.com/software/arcgis/).

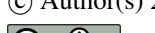



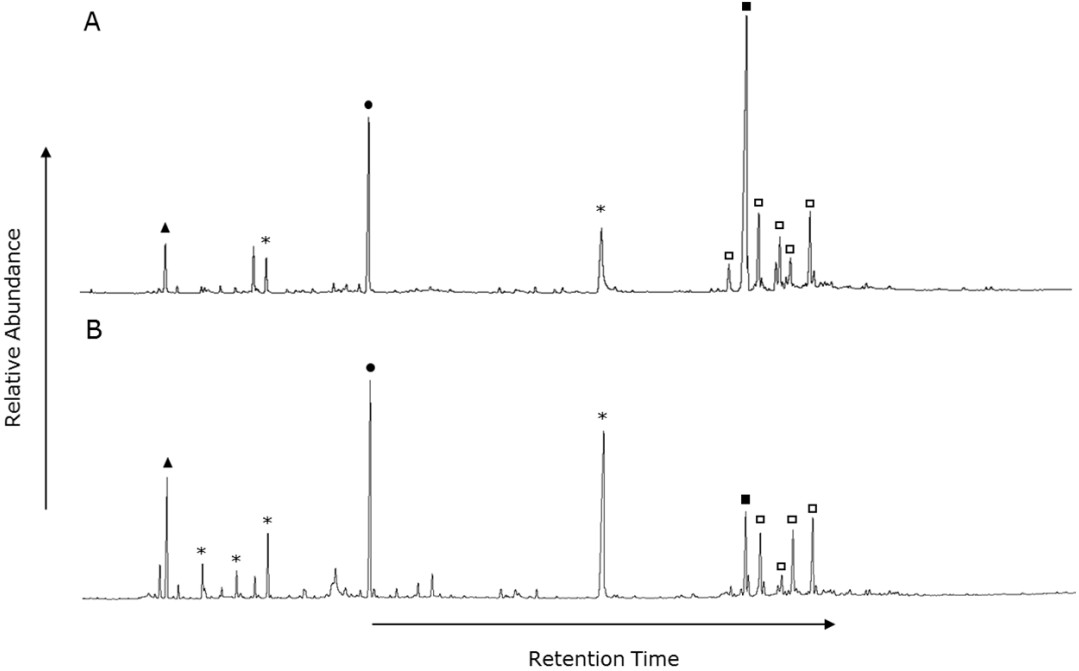

**Figure 2: Chromatogram of silylated polar fraction.** June sediment collected at the A) Control site and B) $CO_2$ vent, showing saturated fatty alcohols (asterisk) and sterols (square), and the representative compounds found among all sample matrices, seasons, and $CO_2$ concentrations: loliolide (triangle), phytol (circle), and cholesterol (closed square).





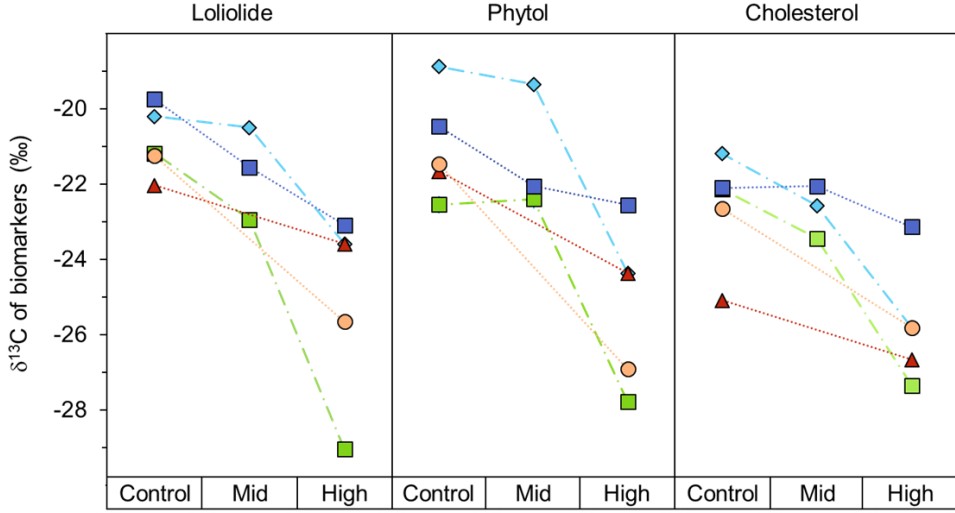

**Figure 3: The δ13C of general algal biomarkers in sediments.** A) Loliolide, B) phytol, and C) cholesterol from the Control, Mid, and High $P$CO₂ sites during June and September from different sample matrices, including surface sediment (square), benthic diatoms (diamond), plankton tow (triangle), and macroalgae (circle).





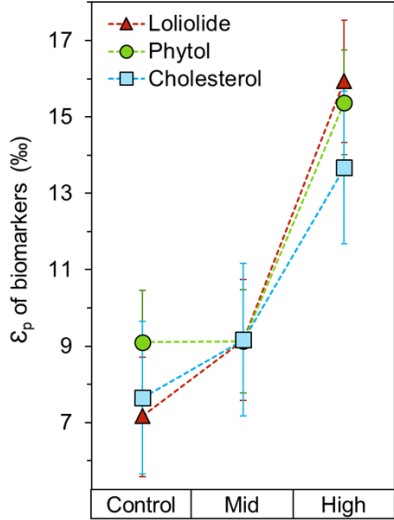

**Figure 4: The ℰ$_p$ of general algal biomarkers in sediments.** Loliolide (triangle), phytol (circle), and cholesterol (square) from the Control, Mid and High $P$CO$_2$ sites during June sediment collection.





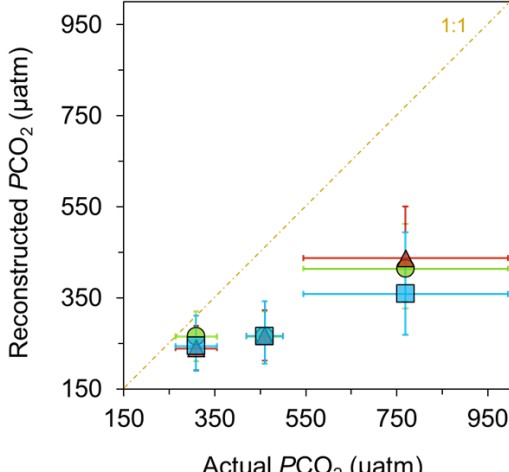

**Figure 5: Reconstructed $PCO_2$ from general algal biomarkers.** $PCO_2$ reconstructed from the $\delta^{13}C$ of loliolide (triangle), phytol (circle), and cholesterol (square) in June-collected sediments versus the actual $PCO_2$ measured at each location (Agostini et al., 2018; Harvey et al. 2018).