# Peer review of "Validation of carbon isotope fractionation in algal lipids as a $PCO_2$ proxy using a natural $CO_2$ seep (Shikine Island, Japan)"

_Biogeosciences, 2019_

## Referee Comment (RC1) · Anonymous Referee #1 · 28 May 2019

Witkoski et al. present a very interesting data-set that suggest that we may be able to use general algal biomarkers for reconstructing past PCO2. My impression of the MS is very positive although it is clear from the results that still more work needs to be done. It is very well organised and easy to read. The date is well presented and the interpretations are sound. I congratulate the authors on their effort. I think that the manuscript should definitely be published in BG and have no major critics. However, I do have a number of specific/technical comments (listed below) that I hope will help to improve the final revision of the MS.

Page 3 Line 9. Suggest placing a reference to figure 1 here.

[Figure]

Page 3 Lines 15-19. Why were the currents and winds measured in 2014 and 2015 and not in 2016 when the samples were collected? How comparable is this for the 'normal' situation in this region?

Page 3 Line 25 Is the abbreviation SPM properly introduced?

Page 4 Line 8. Remove 'then'.

Page 4 Lines 10-11. Change to '. . . . . . NBS-19), flushed with He, injected with 500 $\mu$L of 85% H3PO4, and reacted for 1 h.'

Page 4 Lines 11-12. Change to 'The headspace was measured and average values and standard deviation errors reported are based. . ...'

Page 4 Lines 16-17. Change to '. . ...using ultrasonication (5 times) with 2 ml dichloromethane (DCM): MeOH (9:1 v/v).'

Page 4 Lines 19-20. Change to ' . . ...and the organic matter the DCM layers were pooled and dried over Na2SO4.'

Page 4 Line 20-21. Change to 'The resulting hydrolyzed TLEs were eluted over an alumina packed column and separated into apolar. . ..'

Page 4 Line 22. Remove 'then'

Page 4 Line23-24. Change to '. . ...prior to analyses by gas chromatography-with flame ionization detection (GC-FID), GC-mass spectrometry (GC-MS), and GC- isotope-ratio mass spectrometry (GC-IRMS).'

Page 4 Line 25. Would it not better to report that GC-FID was used for quantification and to check the signal to noise ratio?

Page 4-Line 26. What are the ideal concentrations? What is the range?

Page 4 Line 28. Change to ' . . .. and He is used as carrier gas.'

Page 4 Lines 28-30. Suggest changing it to ' The GC oven was programmed from

70°C to 130°C at 20°C/min and then to 320°C at 4°C/min at which it was held for 10 min. ' Page 4 Line 34. Replace ' C20 and C24' with 'the same'.

Page 5 Lines 6-9. Why include this information again? You have already given this information in the method section.

Page 5 Lines 5-11. If the SPM samples were not included in this study why mention them at all? See no reason for this and suggest removing all information related the SPM samples.

Page 5 Lines 12-13. I cannot find the supplementary information anywhere so cannot comment on this figure.

Page 5 Lines 17-20 +Fig 2. Not all compounds mentioned here are clearly labelled in Fig. 2. For completeness this should be corrected.

Page 5 Line 23 and onwards. Considering that only two (or three) sites are compared it is incorrect to talk about 'change' here (or shift in the next lines). It would be better to report the 'differences' between the sites or, as a couple of lines later, mention if the values are higher or lower if compared to. . ..

Page 6 Line 27– page 7 iine 3. Here the possibility of a contribution of terrestrial derived cholesterol is discussed. I agree that this cannot be completely excluded but was wondering if the authors have some more information about the relative terrestrial contributions to the sediments in this region. Looking at fig 2, for instance, suggest that there is no substantial presence of terrestrial HMW n-alkanols. What about biomarkers present in the other fractions obtained?

Page 7 Lines 15-25. I find this a bit of a confusing section, particularly in line with the information reported in the method section 2.1. As mentioned earlier I do not understand why the currents and winds were measured in 2014 and 2015 and not in 2016. It now seems that the conditions between the sampling seasons were completely abnormal. In addition, would it be possible to add a few references to information given

in this section. I assume that the kind of impact this had on the corals etc must have been properly documented.

Page 8 Line 4. It should be ' Witkowski et al. (2018)'

Page 8 Line 6. Change to '….however, has never been determined.'

Page 8 Line 17. Change to '…...sites for all three….'

Page 8 Line 23. Change to '….as it is the only….'

References. Please check all references carefully. It should be 'Sinninghe Damsté, J. S.' and not 'Damste, J. S. S'.

Figure 3 and 4. Ccurrently the data in these figures is presented as line plots. However, considering that we are only dealing with samples from three discrete areas I feel that this is misrepresenting the results suggesting that there are trend between the three sites. Suggest removing the lines, showing the results as individual points.

---

## Short Comment (SC1) · 11 Jun 2019

I commend Caitlyn Witkowski and the NIOZ group for another impressive paper on paleo-PCO2 reconstruction ("Validation of carbon isotope fractionation in algal lipids as a PCO2 proxy using a natural CO2 seep (Shikine Island, Japan)").

In the last couple of years, some colleagues and collaborators (including Alan Mix) have pointed out to me that the original works of Jasper and Hayes (1990) and subsequently additionally Prahl and Mix (1994) were being overlooked by succeeding researchers in the paleo-PCO2 area. I thank Caitlyn and her colleagues for referencing and integrating our work in this present, important work.

[Figure]

There are a few items that do not seem to translate well into the contemporary paleo-PCO2 literature:

• To my best understanding, Jasper and Hayes (Nature, 1990) was the first to perform specific-molecular paleo-PCO2 reconstruction, in particular, via the insight that sedimentary alkenone d13C record (DSDP 619 from the Gulf of Mexico) could be preliminarily calibrated against an ice core record of pCO2 (the Antarctic Vostok ice core). That was an important advance, connecting the ideas of sedimentary eP to an observed ice-core pCO2 record, opening the opportunity for paleo-PCO2 reconstruction far into the ancient past; and, • This concept was neatly summarized in Figure 3 of Jasper et al. (Paleoceanography, 1994) which has been the template for much succeeding research in this area.

John Hayes and I fully appreciated that the paleo-PCO2 barometer would require further analysis and calibration in contemporary natural and laboratory systems. We were very glad to initially participate in this work (Bidigare, ..Jasper, Hayes et al., Glob. Biogeochem. Cyc., 1997) with many other excellent researchers.

Congratulations on NIOZ's group's excellent work on the very important of PCO2 reconstruction and understanding climate change. As I variously try to impress upon people, global temperature is an exceedingly important climatic variable and pCO2 drives temperature. I hope that the work of Witkowski and colleagues will have influence in the current climate discussion.

---

## Short Comment (SC2) · 28 Jun 2019

We thank John Jasper for reading and reviewing this manuscript. We appreciate the kind words and recommendation for our publication. The work conducted by John Jasper has indeed laid the foundation for the continuing research on isotopic fractionation in algal biomarkers.
* * *

---

## Referee Comment (RC2) · Anonymous Referee #2 · 6 Jul 2019

Witkowski et al. report carbon isotope fractionation from CO2 into algal lipids found in various sample substrates in the vicinity of natural CO2 seeps. They successfully use these sites to ground-truth the use of algal lipid carbon isotope fractionation as a pCO2 proxy. I congratulate the authors on this novel, comprehensive, and concise study. I have some minor comments that should be addressed before acceptance. Further, I would like to ask the authors to use continuous line numbers in the future, as is standard practice.

Line comments: Page 4, 1: Why were the filters combusted only at 300 C for 3h? Standard practice is 450 C for 5h or similar.

[Figure]

Page 5, 6-8: Unclear if the reported pCO2 values (is this dissolved CO2?) are taken from the literature or are original data. If these are original data, the authors need to state in detail how pCO2(aq) was calculated. If these are literature values, and not measured from the same samples as the d13C-DIC values, the authors need to state why they consider these values to be adequate for comparison with their samples (both in a spatial and temporal sense).

Page 5-6: The authors should include all data as either a main text table or supplementary table/data file, containing d13C-DIC, d13C-CO2, d13C of biomarkers etc.

Page 8, 11-12: Is it reasonable to assume a constant temperature? Is there no seasonality in primary productivity at this site?

Page 8, 25-Page 9, 21: Here you could discuss the recent paper by Badger et al. (Climate of the Past, doi. 10.5194/cp-15-539-2019) suggesting insensitivity of alkenone-13C at low-mid pCO2 levels.

Page 9, 18: "annually"

Page 9, 28: I would suggest being more cautious with the wording ("likely") here. Can you provide evidence to support your argument for allochthonous input? Where would this come from?

---

## Short Comment (SC3) · 6 Jul 2019

We thank the reviewer for the comments, as well as for the recommendation for publication. Below, we respond to the specific/technical comments expressed by the reviewer (our response following the *), which will improve the manuscript.

Page 3 Line 9. Suggest placing a reference to figure 1 here.

*We will place a reference to Fig 1 here.

Page 3 Lines 15-19. Why were the currents and winds measured in 2014 and 2015 and not in 2016 when the samples were collected? How comparable is this for the

'normal' situation in this region?

*The currents and winds were studied in detail during the 2014-2015 expeditions (Agostini et al., 2015). Unpublished observational data suggest that this is normal for the region, based on visits to the site on a monthly basis for the past 5 years. Although 2016 was a particularly strong season for storms, this region experiences these kinds of storm activity annually. This annual storm season may explain why the samples collected in this study do not reflect the full PCO2 values observed on site.

Page 3 Line 25 Is the abbreviation SPM properly introduced?

*We will add the appropriate introduction of SPM.

Page 4 Line 8. Remove 'then'.

*We will remove 'then'.

Page 4 Lines 10-11. Change to '...... NBS-19), flushed with He, injected with 500 $\mu$L of 85% H3PO4, and reacted for 1 h.'

*We will change this accordingly.

Page 4 Lines 11-12. Change to 'The headspace was measured and average values and standard deviation errors reported are based.....'

*We will change this accordingly.

Page 4 Lines 16-17. Change to '.....using ultrasonication (5 times) with 2 ml dichloromethane (DCM): MeOH (9:1 v/v).'

*We will change this accordingly.

Page 4 Lines 19-20. Change to ' .....and the organic matter the DCM layers were pooled and dried over Na2SO4.'

*We will change this accordingly.

Page 4 Line 20-21. Change to 'The resulting hydrolyzed TLEs were eluted over an alumina packed column and separated into apolar....'

*We will change this accordingly.

Page 4 Line 22. Remove 'then'

*We will remove 'then'.

Page 4 Line 23-24. Change to '.....prior to analyses by gas chromatography-with flame ionization detection(GC-FID),GC-mass spectrometry (GC-MS), and GC-isotope-ratio mass spectrometry (GC-IRMS).'

*We will change this accordingly.

Page 4 Line 25. Would it not better to report that GC-FID was used for quantification and to check the signal to noise ratio?

*We will change the phrasing of this to the reviewer's recommendation.

Page 4-Line 26. What are the ideal concentrations? What is the range?

*Approximately 50 ng was injected on column.

Page 4 Line 28. Change to ' .... and He is used as carrier gas.'

*We will change this accordingly.

Page 4 Lines 28-30. Suggest changing it to ' The GC oven was programmed from 70C to 130C at 20C/min and then to 320C at 4C/min at which it was held for 10 min. '

*We will change this accordingly.

Page 4 Line 34. Replace ' C20 and C24' with 'the same'.

*We will change this to 'the same'.

Page 5 Lines 6-9. Why include this information again? You have already given this

information in the method section.

*We used this paragraph as a summation of information that was spread across the site and materials subsections of the methods. Based on the reviewer's comments, we will remove this paragraph.

Page 5 Lines 5-11. If the SPM samples were not included in this study why mention them at all? See no reason for this and suggest removing all information related the SPM samples.

*This information was included to let readers know that SPM was considered, though the clear waters did not yield enough material. We can minimize the reference to SPM to a single sentence within the manuscript.

Page 5 Lines 12-13. I cannot find the supplementary information anywhere so cannot comment on this figure.

*Thank you for pointing this out. We will upload the supplementary material.

Page 5 Lines 17-20 and Fig 2. Not all compounds mentioned here are clearly labelled in Fig. 2. For completeness this should be corrected.

*We have not specifically indicated all compounds as they crowd the chromatograms and are not part of the study. Our point was merely to show that the compounds investigated are abundant and that there are no large differences between sites.

Page 5 Line 23 and onward. Considering that only two (or three) sites are compared it is incorrect to talk about 'change' here (or shift in the next lines). It would be better to report the 'differences' between the sites or, as a couple of lines later, mention if the values are higher or lower if compared to....

*We will change this phrasing to 'differences' from 'change'.

Page 6 Line 27– page 7 line 3. Here the possibility of a contribution of terrestrial derived cholesterol is discussed. I agree that this cannot be completely excluded but

was wondering if the authors have some more information about the relative terrestrial contributions to the sediments in this region. Looking at fig 2, for instance, suggest that there is no substantial presence of terrestrial HMW n-alkanols. What about biomarkers present in the other fractions obtained?

*As seen in this Fig 2, these samples lacked triterpenoids and long-chain alcohols typical of higher plants, suggesting lack of terrestrial input. Furthermore, our apolar fractions lacked long-chain n-alkanes associated with terrestrial input. Therefore, we think we can exclude this as a possible issue with cholesterol. This will be noted in the revised manuscript.

Page 7 Lines 15-25. I find this a bit of a confusing section, particularly in line with the information reported in the method section 2.1. As mentioned earlier I do not understand why the currents and winds were measured in 2014 and 2015 and not in 2016. It now seems that the conditions between the sampling seasons were completely abnormal. In addition, would it be possible to add a few references to information given in this section. I assume that the kind of impact this had on the corals etc must have been properly documented.

*Based on unpublished observational data from part of the co-authors visiting the site on a monthly basis, the conditions covering 2014-2015 are typical of what they've witnessed over the past six years of research. This study on winds and currents was intensely time-consuming and was thus not repeated here. However, we will add weather station data to further support the normalness of currents and winds in this region of Japan (unfortunately, the only recorded data from this specific island is referenced here). As commented above, typhoons and strong tropical storms are common in this region and occur on an annual basis. Our June collected samples showed a lessened reconstructed value for PCO2 than the PCO2 measured at the site, which may be explained by this annual storm season which mixes the bay every year. Mixing during the storm season is a plausible explanation to why we observe a distinct difference between our two sampling seasons (one before the storm season and one after). Our

sampling year (2016) happened to have particularly intense storms reaching land (see supporting figure attached here).

Page 8 Line 4. It should be ' Witkowski et al. (2018)'

*This will be changed to Witkowski et al. (2018).

Page 8 Line 6. Change to '....however, has never been determined.'

*This will be changed to '…however, has never been determined.'

Page 8 Line 17. Change to '.....sites for all three....'

*This will be changed to 'for'.

Page 8 Line 23. Change to '....as it is the only....'

*This will be changed to 'as it is the only'

References. Please check all references carefully. It should be 'Sinninghe Damsté, J. S.' and not 'Damste, J. S. S'.

*We will check the references more carefully.

Figure 3 and 4. Currently the data in these figures is presented as line plots. However, considering that we are only dealing with samples from three discrete areas I feel that this is misrepresenting the results suggesting that there are trend between the three sites. Suggest removing the lines, showing the results as individual points.

*We will remove the lines and show the results as individual points.
* * *
**Typhoons by Year**

Typhoons generated

Typhoons making landfall in Japan

Created by *Nippon.com* based on data from the Japan
Meteorological Agency (as of July 2018).

nippon.com

**Fig. 1.**

---

## Short Comment (SC4) · 11 Jul 2019

*We thank the reviewer for the comments and recommendation for publication. Below we will respond to each of the comments (italics), which will improve the manuscript.*

Line comments: . . .I would like to ask the authors to use continuous line numbers in the future, as is standard practice.

*We used the Biogeosciences format which specifies using the numbering shown in this manuscript (though we also prefer continuous line numbering).*

[Figure]

Page 4, 1: Why were the filters combusted only at 300 C for 3h? Standard practice is 450 C for 5h or similar.

***This temperature is sufficient to remove all background molecules which potentially can contribute the lipid pool we investigated. Indeed, higher temperatures are needed in case one wants to have a completely carbon-free filter.***

Page 5, 6-8: Unclear if the reported pCO2 values (is this dissolved CO2?) are taken from the literature or are original data. If these are original data, the authors need to state in detail how pCO2(aq) was calculated. If these are literature values, and not measured from the same samples as the d13C-DIC values, the authors need to state why they consider these values to be adequate for comparison with their samples (both in a spatial and temporal sense).

***This data is reported in other studies (e.g. Agostini et al., 2015; 2018; Harvey et al., 2018) where it is specified how they came to these values, i.e. calculated PCO2 based on the carbonate chemistry parameters of the bay (using the program CO2sys). We will briefly expand on this section.***

Page 5-6: The authors should include all data as either a main text table or supplementary table/data file, containing d13C-DIC, d13C-CO2, d13C of biomarkers etc.

***We will include a main text table with the values used to reconstruct PCO2 at this site.***

Page 8, 11-12: Is it reasonable to assume a constant temperature? Is there no seasonality in primary productivity at this site?

***Here, we use a constant temperature because the surface sediments are an integrated accumulation of all primary productivity over the year. Although primary productivity is higher in the spring and summer, this site has some (observational) productivity throughout the year.***

Page 8, 25-Page 9, 21: Here you could discuss the recent paper by Badger et al.

(Climate of the Past, doi. 10.5194/cp-15-539-2019) suggesting insensitivity of alkenone 13C at low-mid pCO2 levels.

***Badger et al. shows an insensitivity to the alkenone proxy at low CO2 values (<400 μatm), but here our general biomarkers do reconstruct the correct (low) control values and rather show insensitivity to the higher PCO2 sites. We will briefly discuss this citation in the revised manuscript.***

Page 9, 18: "annually"

***This will be changed to "annually"***

Page 9, 28: I would suggest being more cautious with the wording ("likely") here. Can you provide evidence to support your argument for allochthonous input? Where would this come from?

***We use the word "likely" here as we do not have independent support for the input of allochthonous organic matter. That material would come from surface sediments transported from the edge or outside of the bay where CO2 levels are much lower than near the CO2 seep. Since this is not a very large distance (500 meters) we can imagine that strong circulation events like typhoons would resuspend surface sediments and transport them to near CO2 vents.***

---

## Referee Comment (RC3) · Anonymous Referee #2 · 12 Jul 2019

I have two comments that warrant further clarification by the authors:

Regarding combustion of filters: The authors should provide literature evidence for their statement that the target compounds would be completely degraded after 3h at 300 °C. I have a hard time believing especially that phytol would combust to completion under these conditions (which are not too different from the GC conditions used by the authors).

Regarding literature pCO2 values: After perusal of Agostini et al. (2015, 2018) and Harvey et al. (2018) it is still not clear to me how the authors derived at the presented pCO2 values. The pCO2 data presented in the former studies show high variability

both temporally and spatially, which should not be disregarded in the present study.

---

## Short Comment (SC5) · 15 Jul 2019

*We thank the reviewer for their additional comments, which we hope we address in the following response.*

I have two comments that warrant further clarification by the authors: Regarding combustion of filters: The authors should provide literature evidence for their statement that the target compounds would be completely degraded after 3h at 300C. I have a hard time believing especially that phytol would combust to completion under these conditions (which are not too different from the GC conditions used by the authors).

[Figure]

*We thank the reviewer for bringing this to our attention. Upon further investigation, we have found that the standard procedure used in our lab (and thus used in this study) was 450C for 4 h. Our apologies for the confusion.*

Regarding literature pCO2 values: After perusal of Agostini et al. (2015, 2018) and Harvey et al. (2018) it is still not clear to me how the authors derived at the presented pCO2 values. The pCO2 data presented in the former studies show high variability both temporally and spatially, which should not be disregarded in the present study.

*The PCO2 data presented in the former studies were calculated using the carbonate chemistry system analysis program CO2SYS using the measured values for pHNBS, temperature, salinity, and total alkalinity (TA) values. There is indeed high variability both temporally and spatially. On Page 3, Line 11-12, we include the standard deviations (Control PCO2 309 $\pm$ 46 $\mu$atm, Mid PCO2 ca. 460 $\pm$ 40 $\mu$atm, and High PCO2 769 $\pm$ 225 $\mu$atm). In Figure 5, these standard deviations are included as horizontal error bars where the "Actual PCO2" values measured at the site lie on the x-axis.*

*Based on the reviewer's comment, we recognize the need to emphasize and discuss this measured high variability at the sites. This variability could have major impacts on the reconstructed values, as these algae are exposed to different levels of PCO2 even within the same site.*

---

## Referee Comment (RC4) · Anonymous Referee #3 · 14 Aug 2019

The authors of the manuscript use natural CO2 seeps in the vicinity of Shikine Island (Japan) to investigate the relationship between different concentrations of aqueous pCO2 and carbon isotope fractionation in three organic compounds extracted from surface marine sediments, diatoms, plankton tow, and microalgae. It is a novel approach that utilizes a unique natural setting. The subject of the manuscript fits well within the scope of the journal, and the results of this project would certainly be of significant interest to paleoceanographers and paleoclimatologists who use carbon isotopic composition of biomarkers as a proxy for pCO2. The manuscript, however, contains several major and minor issues that need to be addressed before the manuscript is considered for publication.

[Figure]

**MAJOR ISSUES**

FIRST, the choice of organic compounds (biomarkers) The authors need to provide a clear rationale as to why loliolide, phytol and cholesterol were chosen for this work. None of these compounds can uniquely be linked with a source (i.e. they can come from a variety of sources including marine and terrestrial), so it is not clear how applicable their work (assuming these compounds are targeted) would be to paleo studies. In fact, the problem with significant underestimation of reconstructed pCO2 (see the next issue below) might be due to a poor control of what those compounds actually represent in terms of the source in this study.

SECOND, underestimation of reconstructed pCO2 Figure 5 and the accompanying discussion show that the reconstructed pCO2 are significantly lower than the measured values at both the Mid and High pCO2 sites by almost a factor of two. The possible reason(s) for this are not really addressed and mainly limited to "some site limitation". This issue requires a more detailed discussion particularly with regard to possible influences of different OM sources and the validity of the assumptions used for calculating the epsilon values for each compound (Section 4.2).

**MINOR ISSUES**

p. 1, line 16, "general algal compounds": What does this mean? Are these compounds sources only by algae?

p. 2, line 1, "current proxies leave much to be desired, often with large uncertainties and conflicting values": Could the authors elaborate on this, i.e. what specific issues with the current proxies do the authors have in mind and how this work would reduce these limitations?

p. 3, line 25, "SPM": What does SPM stand for?

p. 6, line 14, "lighter d13C values": a d13C value cannot be 'lighter' or 'heavier'. It is a number. Use 'lower' or 'higher' instead.

p. 6, lines 18-19, 29-30, "the primarily diatom-limited compound loliolide": It is a very common compound derived from many sources, including macrophytic algae and terrestrial plants, so linking it specifically with diatoms is somewhat risky. Furthero-more, this compound is know to be a degradation product of fucoxanthin and other carotenoids, whick are also difficult to link to a particular source during paleorecon-structions.

p. 9, lines17-18, "allochthonous input of sediment": Need to provide more detail here. Is it just about sediment or about are organic matter/biomarker sources with different epsilon values that would make reconstructing pCO2 more complex?

FIGURES

Figure1: The figure is confusing, i.e. it is difficult to know where this island is. It needs to be shown in a broader context, e.g. with a map of Japan at least. Geographic maps also typically have lines of latitude and longitude (shown as grid) along the X- and Y-axes. Also, the direction of the geographic North should be indicated.

Figure 2: Is it a GC-FID trace or GC-MS (TIC or SI mode, if so which m/z?)? Why not to give the names of the compounds next to the peaks rather than list them in the caption?

Figure 3: A), B), and C) are not shown on the plots. These need to be labelled. What are the error bars associated with the d13C values shown on the plots? Also, instead of 'Control', 'Mid', and 'High' show the actual pCO2 values.

Figure 4: Here and in text (p. 8, lines 18-20), explain how the errors associated with the epsilon(p) values were calculated?

—————————————————————

---

## Short Comment (SC6) · 16 Aug 2019

**Response to RC4 comments**

*The authors of the manuscript use natural CO2 seeps in the vicinity of Shikine Island(Japan) to investigate the relationship between different concentrations of aqueouspCO2 and carbon isotope fractionation in three organic compounds extracted from surface marine sediments, diatoms, plankton tow, and microalgae. It is a novel approach that utilizes a unique natural setting. The subject of the manuscript fits well within the scope of the journal, and the results of this project would certainly be of significant interest to paleoceanographers and paleoclimatologists who use carbon isotopic*

*composition of biomarkers as a proxy for pCO2. The manuscript, however, contains several major and minor issues that need to be addressed before the manuscript is considered for publication.*

**We thank the reviewer for the comments and recommendation for publication. Below we will respond to each of the comments, which will improve the manuscript.**

*MAJOR ISSUES*

*FIRST, the choice of organic compounds (biomarkers). The authors need to provide a clear rationale as to why loliolide, phytol and cholesterol were chosen for this work. None of these compounds can uniquely be linked with a source (i.e. they can come from a variety of sources including marine and terrestrial), so it is not clear how applicable their work (assuming these compounds are targeted) would be to paleo studies. In fact, the problem with significant underestimation of reconstructed pCO2 (see the next issue below) might be due to a poor control of what those compounds actually represent in terms of the source in this study.*

**We chose these organic compounds because they are representative of a wide range of producers, the concept being to offer a complementary approach to species-specific compounds (i.e. alkenones) that are temporarily and spatially limited. By exploring a larger groups of producers in open ocean settings, we may be able to extend the PCO2 record derived from epsilon p, as has been done for the Cretaceous (Bice et al., 2006; Sinninghe Damsté et al., 2008; Naafs et al., 2016) and for the Phanerozoic (Witkowski et al., 2018), both reconstructed from phytol's diagenetic product phytane. Although the sources of these compounds may be both terrestrial and/or marine, when viewed in an open marine setting will almost entirely be from phytoplankton (and in the case of cholesterol, also the zooplankton that consume and retain the isotopic composition of these same phytoplankton). Here, we are on the coast of a small island in open ocean and**

**have scanned our chromatogram for characteristic terrestrial biomarkers to test whether the contributors of these compounds also include terrestrial inputs from the island. The lack of triterpenoids and long-chain alcohols typical of higher plants suggests that our source signal is overwhelmingly marine.**

*SECOND, underestimation of reconstructed pCO2 Figure 5 and the accompanying discussion show that the reconstructed pCO2 are significantly lower than the measured values at both the Mid and High pCO2 sites by almost a factor of two. The possible reason(s) for this are not really addressed and mainly limited to "some site limitation". This issue requires a more detailed discussion particularly with regard to possible influences of different OM sources and the validity of the assumptions used for calculating the epsilon values for each compound (Section 4.2).*

**In section 4.3, we reconstruct PCO2 and describe why these reconstructed values are lower than the measured high PCO2 sites, primarily focused on the novelty of using such a site and the further research required. We have discounted different OM sources due to the lack of terrestrial biomarkers.**

**However, we agree with the referee and will expand on several sections to further consider the criticisms of epsilon p. First, we will expand the end of 4.2 to include why epsilon f (maximum fractionation) is not fully expressed at the high CO2 site, such as species' affinity for carbon concentration mechanisms which utilize 13C-enriched bicarbonate, as well as the recent studies that show different Rubisco types may yield lower epsilon f than previously assumed (Thomas et al., 2018). Second, we will expand the end of section 4.3 to raise the possibility of changing b value (factors influencing fractionation other than CO2) which has been shown to vary (Zhang et al., 2019).**

*MINOR ISSUES p. 1, line 16, "general algal compounds": What does this mean? Are these compounds sources only by algae?*

**As per our response in major issues 1, general algal compounds contrast to**

**species-specific algal biomarkers, i.e. alkenones. General algal biomarkers refer to compounds that are derived from a multitude of species, presumably overwhelmingly from phytoplankton sources based on our analyses. We will clarify this in the text.**

*p. 2, line 1, "current proxies leave much to be desired, often with large uncertainties and conflicting values": Could the authors elaborate on this, i.e. what specific issues with the current proxies do the authors have in mind and how this work would reduce these limitations?*

**We will elaborate on this. Although there has been much progress in development of PCO2 proxies, there are few proxies which can span timescales over 100 Ma. The few that can span longer periods are terrestrial biomarkers, which tend to have larger uncertainties, e.g. paleosols. Epsilon p has its problems, particularly at lower PCO2 but tends to have smaller uncertainties and so if this could be applied to longer timescales, it would offer a marine record (less influenced by local carbon cycling) and could help constrain the estimates for these older records.**

*p. 3, line 25, "SPM": What does SPM stand for?*

**Another reviewer also pointed this out. We have now defined this as suspended particulate matter the first time this is mentioned.**

*p. 6, line 14, "lighter d13C values": a d13C value cannot be 'lighter' or 'heavier'. It is a number. Use 'lower' or 'higher' instead.*

**We will change this to higher (13C enrichment) or lower (13C depletion) throughout the manuscript.**

*p. 6, lines 18-19, 29-30, "the primarily diatom-limited compound loliolide": It is an very common compound derived from many sources, including macrophytic algae and terrestrial plants, so linking it specifically with diatoms is somewhat risky. Furthero-*

*more, this compound is know to be a degradation product of fucoxanthin and other carotenoids, whick are also difficult to link to a particular source during paleoreconstructions.*

**We will add a small section to further describe the sources and, as per Major Issues 1, describe why we chose these specific compounds.**

**Regarding the source of loliolide, it is established that it is a product primarily from fucoxanthin. Repeta (1988) explores possible carotenoid sources of loliolide in modern sediments and demonstrate the fucoxanthin contributes to loliolide but are unable to demonstrate a parallel conversion of diadinoxanthin and other carotenoid epoxides to loliolide. Fucoxanthin is found in diatoms, as well as brown seaweeds, and is not common in terrestrial plants. The vast majority of fucoxanthin in the world is derived from diatoms, which make up a vastly larger mass of producers than brown seaweeds and generally contain more than four times as much fucoxanthin as brown seaweeds.**

**We will add a sentence to further describe the different possible sources of loliolide. However, given that all the theoretical sources are carotenoids, these should have the same biosynthetic pathways to be produced and thus should not affect the isotopic composition of the degradation product loliolide.**

*p. 9, lines17-18, "allochthonous input of sediment": Need to provide more detail here. Is it just about sediment or about are organic matter/biomarker sources with different epsilon values that would make reconstructing pCO2 more complex?*

**We will add more detail here to describe what we mean by allochthonous input, here referring to the deposit of sediment that contain our organic compounds that has originated at a distance (e.g. the control) into our elevated PCO2 sites. Sediment mixed between the high PCO2 site and the control site would likewise mix the epsilon p signal derived from these sediments.**

*FIGURES Figure1: The figure is confusing, i.e. it is difficult to know where this island is. It needs to be shown in a broader context, e.g. with a map of Japan at least. Geographic maps also typically have lines of latitude and longitude (shown as grid) along the X- and Y-axes. Also, the direction of the geographic North should be indicated.*

**We will revise this map to include an insert of the larger region (i.e. Japan) with the location of the island. We will also include latitude and longitude lines on the x- and y-axis, as well as geographic North.**

*Figure 2: Is it a GC-FID trace or GC-MS (TIC or SI mode, if so which m/z?)? Why not to give the names of the compounds next to the peaks rather than list them in the caption?*

**This is an GC-FID trace, which we will label. We will put the compounds next to the peaks rather than in the caption.**

*Figure 3: A), B), and C) are not shown on the plots. These need to be labelled. What are the error bars associated with the d13C values shown on the plots? Also, instead of 'Control', 'Mid', and 'High' show the actual pCO2 values.*

**We will add labels for A, B, and C. The error bars are 0.5‰ as described in the text. These were difficult to see in the figure, as they all overlap with one another. We will add these in for the referee. We will add the actual PCO2 values.**

*Figure 4: Here and in text (p. 8, lines 18-20), explain how the errors associated with the epsilon(p) values were calculated?*

**We thank the referee for pointing this out (especially as this was quite time-consuming, and we did not explain it!). We will add a section on how the uncertainties are calculated for both epsilon p and PCO2, which show one standard deviation (68 percent) uncertainty in based on Monte Carlo simulations, culminating the uncertainty in every equation parameter outlined in this manuscript.**

---

## Author Comment (AC1) · 13 Sep 2019

For our response, see short comment

---

## Author Comment (AC2) · 13 Sep 2019

Four our response, see short comment

---

## Author Comment (AC3) · 13 Sep 2019

for our response, see short comment
* * *

---

## Author Comment (AC4) · 13 Sep 2019

for our response, see short comment

————————————————

---

## Author Comment (AC5) · 13 Sep 2019

for our rresponse, see short comment

---

## Author Response (AR2)

10 September 2019

**RE: Revisions to manuscript**

Dear Editor, Dr. Woulds,

Please find attached our revised submission to *Biogeosciences* entitled "**Validation of carbon isotope fractionation in algal lipids as a $P$CO$_2$ proxy using a natural CO$_2$ seep (Shikine Island, Japan)**" by Caitlyn R. Witkowski, Sylvain Agostini, Ben P. Harvey, Marcel T.J. van der Meer, Jaap S. Sinninghe Damsté, and Stefan Schouten. We would like to thank the editor and reviewers for their thoughtful and constructive criticism, which has resulted in an improved manuscript.

In the comments by the reviewers, most issues were primarily on details or technical. Some of these comments were the need to include more information or clarification on the site conditions (e.g. occurrence of extreme weather, typical temperatures, $P$CO$_2$ measurements). A major comment which we addressed is the other potential influences on the $P$CO$_2$ reconstructed from these potential proxies. We hope that we have adequately addressed their concerns, which we do in detail below.

Sincerely,
On behalf of all co-authors,

**Caitlyn R. Witkowski**
caitlyn.witkowski@bristol.ac.uk
Postdoctoral Research Associate
School of Earth Sciences
University of Bristol

**Detailed Response to Reviewer comments**

*Reviewer 1*
*Witkowski et al. present a very interesting dataset that suggest that we may be able to use general algal biomarkers for reconstructing past PCO2. My impression of the MS is very positive although it is clear from the results that still more work needs to be done. It is very well organised and easy to read. The date is well presented and the interpretations are sound. I congratulate the authors on their effort. I think that the manuscript should definitely be published in BG and have no major critics. However, I do have a number of specific/technical comments (listed below) that I hope will help to improve the final revision of the MS.*

**We thank the reviewer for the comments, as well as for the recommendation for publication. Below, we respond to the specific/technical comments expressed by the reviewer.**

*Page 3 Line 9. Suggest placing a reference to figure 1 here.*

**We have placed a reference to Fig 1 here.**

*Page 3 Lines 15-19. Why were the currents and winds measured in 2014 and 2015 and not in 2016 when the samples were collected? How comparable is this for the 'normal' situation in this region?*

**The currents and winds were studied in detail during the 2014-2015 expeditions (Agostini et al., 2015). Unpublished observational data suggest that the observed currents and winds are normal for the region, based on visits to the site on a monthly basis for the past 5 years. Although 2016 was a particularly strong season for storms, this region experiences these kinds of storm activity annually.**

*Page 3 Line 25. Is the abbreviation SPM properly introduced?*

**We have removed all mention of SPM, as these samples did not yield enough organic material for isotopic measurements.**

*Page 4 Line 8. Remove 'then'.*

**We have removed 'then'.**

*Page 4 Lines 10-11. Change to '...... NBS-19), flushed with He, injected with 500 µL of 85% H3PO4, and reacted for 1 h.'*

**We have changed this phrasing.**

*Page 4 Lines 11-12. Change to 'The headspace was measured and average values and standard deviation errors reported are based....'*

**We have changed this phrasing.**

*Page 4 Lines 16-17. Change to '.... using ultrasonication (5 times) with 2 ml dichloromethane (DCM): MeOH (9:1 v/v).'*

**We have changed this phrasing.**

*Page 4 Lines 19-20. Change to '.....and the organic matter the DCM layers were pooled and dried over Na2SO4.'*

**We have changed this phrasing.**

*Page 4 Line 20-21. Change to 'The resulting hydrolyzed TLEs were eluted over an alumina packed column and separated into apolar....'*

**We have changed this phrasing.**

*Page 4 Line 22. Remove 'then'*

**We have removed 'then'.**

*Page 4 Line23-24. Change to '.....prior to analyses by gas chromatography-with flame ionizationdetection(GC-FID),GC-massspectrometry(GC-MS),andGC-isotope-ratio mass spectrometry (GC-IRMS).'*

**We have changed this to "analyses".**

*Page 4 Line 25. Would it not better to report that GC-FID was used for quantification and to check the signal to noise ratio?*

**We have changed this phrasing.**

*Page 4-Line 26. What are the ideal concentrations? What is the range?*

**We have added that ca. 1 ug of polar fraction was injected on-column.**

*Page 4 Line 28. Change to '.... and He is used as carrier gas.'*

**We have changed this phrasing.**

*Page 4 Lines 28-30. Suggest changing it to 'The GC oven was programmed from70◦C to 130◦C at 20◦C/min and then to 320◦C at 4◦C/min at which it was held for 10 min.'*

**We have changed this phrasing.**

*Page 4 Line 34. Replace 'C20 and C24' with 'the same'.*

**We have changed this to 'the same'.**

*Page 5 Lines 6-9. Why include this information again? You have already given this information in the method section.*

**We used this paragraph as a summation of information that was spread across the site and materials subsections of the methods. Although we agree with the reviewer that there is some repetition, we think it is important to include in both sections.**

*Page 5 Lines 5-11. If the SPM samples were not included in this study why mention them at all? See no reason for this and suggest removing all information related the SPM samples.*

**We have now removed all mention of SPM.**

*Page 5 Lines 12-13. I cannot find the supplementary information anywhere so cannot comment on this figure.*

**Thank you for pointing this out. We have now included the supplementary material.**

*Page 5 Lines 17-20 +Fig 2. Not all compounds mentioned here are clearly labelled in Fig. 2. For completeness this should be corrected.*

**We have not specifically indicated all compounds as they crowd the chromatograms and are not part of the study. Our point was merely to show that the compounds investigated are abundant, well-separated, and that there are no large differences between sites.**

*Page 5 Line 23 and onwards. Considering that only two (or three) sites are compared it is incorrect to talk about 'change' here (or shift in the next lines). It would be better to report the 'differences' between the sites or, as a couple of lines later, mention if the values are higher or lower if compared to....*

**We have changed this phrasing to 'differences' from 'change'.**

*Page 6 Line 27– page 7 iine 3. Here the possibility of a contribution of terrestrial derived cholesterol is discussed. I agree that this cannot be completely excluded but was wondering if the authors have some*

*more information about the relative terrestrial contributions to the sediments in this region. Looking at fig 2, for instance, suggest that there is no substantial presence of terrestrial HMW n-alkanols. What about biomarkers present in the other fractions obtained?*

**We have added a sentence here that explains that the samples lacked triterpenoids and long-chain alcohols typical of higher plants, suggesting a low amount of terrestrial input.**

*Page 7 Lines 15-25. I find this a bit of a confusing section, particularly in line with the information reported in the method section 2.1. As mentioned earlier I do not understand why the currents and winds were measured in 2014 and 2015 and not in 2016. It now seems that the conditions between the sampling seasons were completely abnormal. In addition, would it be possible to add a few references to information given in this section. I assume that the kind of impact this had on the corals etc must have been properly documented.*

**Based on unpublished observational data from part of the co-authors visiting the site on a monthly basis, the conditions covering 2014-2015 are typical of what they've witnessed over the past six years of research. This study on winds and currents is intensely time-consuming and was thus not repeated in 2016.**

**As commented above, typhoons and strong tropical storms are common in this region and occur on an annual basis. Our June collected samples showed lower reconstructed values for $P$CO$_2$ than September, which may be explained by this annual storm season which mixes the surface sediments of the bay every year. Our sampling year (2016) happened to have particularly intense storms reaching land (see supporting figure attached here).**

**We have added a sentence in Section 2.1 that states, "Monthly surveys in the bays over the past five years show that these sites have similar annual mean values for temperature, salinity, and currents. Weather station data shows that severity of seasonal extreme weather, e.g. typhoons, varies on an annual basis (Japan Meteorological Agency)."**

[Figure]

**Typhoons by Year**

Created by *Nippon.com* based on data from the Japan Meteorological Agency (as of July 2018).

*Page 8 Line 4. It should be ' Witkowski et al. (2018)'*

**This is changed to Witkowski et al. (2018).**

*Page 8 Line 6. Change to '....however, has never been determined.'*

**This is changed to '…however, has never been determined.'**

*Page 8 Line 17. Change to '.....sites for all three....'*

**This is changed to 'for'.**

*Page 8 Line 23. Change to '....as it is the only....'*

**This is changed to 'as it is the only'**

*References. Please check all references carefully. It should be 'Sinninghe Damsté, J. S.' and not 'Damste, J. S. S'.*

**We have checked the references more carefully.**

*Figure 3 and 4. Currently the data in these figures is presented as line plots. However, considering that we are only dealing with samples from three discrete areas I feel that this is misrepresenting the results suggesting that there are trend between the three sites. Suggest removing the lines, showing the results as individual points.*

**We have removed the lines and show the results as individual points.**

*Reviewer 2:*

*Witkowski et al. report carbon isotope fractionation from CO2 into algal lipids found in various sample substrates in the vicinity of natural CO2 seeps. They successfully use these sites to ground-truth the use of algal lipid carbon isotope fractionation as a pCO2 proxy. I congratulate the authors on this novel, comprehensive, and concise study. I have some minor comments that should be addressed before acceptance. Furthermore, I would like to ask the authors to use continuous line numbers in the future, as is standard practice.*

**We thank the reviewer for the recommendation for publication and comments which have improved the manuscript. The reviewer had asked us in a separate comment for a clarification on two comments in our rebuttal, which we have included under the corresponding original questions (Comments on Page 4, 1 and on Page 5, 6-8).**

*Line comments:*

*…I would like to ask the authors to use continuous line numbers in the future, as is standard practice.*

**We used the Biogeosciences format which specifies using the numbering shown in this manuscript (though we also prefer continuous line numbering).**

*Page 4, 1: Why were the filters combusted only at 300 C for 3h? Standard practice is 450 C for 5h or similar.*

**We thank the reviewer for bringing this to our attention. Upon further investigation, we have found that the standard procedure used in our lab (and thus used in this study) was 450∘C for 4 h. Our apologies for the confusion.**

*Page 5, 6-8: Unclear if the reported pCO2 values (is this dissolved CO2?) are taken from the literature or are original data. If these are original data, the authors need to state in detail how pCO2(aq) was calculated. If these are literature values, and not measured from the same samples as the d13C-DIC values, the authors need to state why they consider these values to be adequate for comparison with their samples (both in a spatial and temporal sense).*

**The $P\text{CO}_2$ data presented in the former studies were calculated using the carbonate chemistry system analysis program CO2SYS using the measured values for $pH_{NBS}$, temperature, salinity, and total alkalinity (TA) values. There is indeed high variability both temporally and spatially. On Page 3, Line 11-12, we include the standard deviations (Control $P\text{CO}_2$ 309 ± 46 µatm, Mid $P\text{CO}_2$ ca. 460 ± 40 µatm, and High $P\text{CO}_2$ 769 ± 225 µatm). In Figure 5, these standard deviations are included as horizontal error bars where the "Actual $P\text{CO}_2$" values measured at the site lie on the x-axis.**

**Based on the reviewer's comment, we have now added a few lines to the discussion in 4.3 regarding possible cause for $P\text{CO}_2$ underestimation in our mid and high sites, in which we state that this variability could have major impacts on the reconstructed values, as these algae are exposed to different levels of $P\text{CO}_2$ even within the same site.**

*Page 5-6: The authors should include all data as either a main text table or supplementary table/data file, containing d13C-DIC, d13C-CO2, d13C of biomarkers etc.*

**We have included three supplementary tables, one with $\delta^{13}\text{C}$ of all the measurements taken, one of the calculations used to derive a corrected $\delta^{13}\text{C}$ of $\text{CO}_2$, and one with $\varepsilon_p$ calculations to estimated $P\text{CO}_2$ for all three biomarkers.**

*Page 8, 11-12: Is it reasonable to assume a constant temperature? Is there no seasonality in primary productivity at this site?*

**Here, we use an annual average temperature because the surface sediments are an integrated accumulation of all primary productivity over the year. Although primary productivity is higher in the spring and summer, this site has some (observational) productivity throughout the year.**

*Page 8, 25-Page 9, 21: Here you could discuss the recent paper by Badger et al. (Climate of the Past, doi. 10.5194/cp-15-539-2019) suggesting insensitivity of alkenone13C at low-mid pCO2 levels.*

**In the discussion 4.3, we have added "There are several possible explanations to why there is an underestimation. As discussed above, carbonate concentration mechanisms may be operating in a large number of phytoplankton, such that they become relatively enriched in 13C and thus lead to lower reconstructed PCO2 values (Badger et al., 2019; Stoll et al., 2019)." We have also added the possibility of a variable b value.**

*Page 9, 18: "annually"*

**This is changed to "annually"**

*Page 9, 28: I would suggest being more cautious with the wording ("likely") here. Can you provide evidence to support your argument for allochthonous input? Where would this come from?*

**We use the word "likely" here as we do not have independent support for the input of allochthonous organic matter. That material would come from surface sediments transported from the edge or outside of the bay where $\text{CO}_2$ levels are much lower than near the $\text{CO}_2$ seep. Since this is not a very large distance (500 meters) we can imagine that strong circulation events like typhoons**

would resuspend surface sediments and transport them to near $CO_2$ vents. In the conclusion, we have added, "from nearby sediments deposited under normal $P\text{CO}_2$ levels caused by the intense annual typhoon activity in this region."

*Reviewer 3:*

*The authors of the manuscript use natural CO2 seeps in the vicinity of Shikine Island(Japan) to investigate the relationship between different concentrations of aqueouspCO2 and carbon isotope fractionation in three organic compounds extracted from surface marine sediments, diatoms, plankton tow, and microalgae. It is a novel approach that utilizes a unique natural setting. The subject of the manuscript fits well within the scope of the journal, and the results of this project would certainly be of significant interest to paleoceanographers and paleoclimatologists who use carbon isotopic composition of biomarkers as a proxy for pCO2. The manuscript, however, contains several major and minor issues that need to be addressed before the manuscript is considered for publication.*

**We thank the reviewer for the comments and recommendation for publication. Below we will respond to each of the comments, which will improve the manuscript.**

*MAJOR ISSUES*
*FIRST, the choice of organic compounds (biomarkers). The authors need to provide a clear rationale as to why loliolide, phytol and cholesterol were chosen for this work. None of these compounds can uniquely be linked with a source (i.e. they can come from a variety of sources including marine and terrestrial), so it is not clear how applicable their work (assuming these compounds are targeted) would be to paleo studies. In fact, the problem with significant underestimation of reconstructed pCO2 (see the next issue below) might be due to a poor control of what those compounds actually represent in terms of the source in this study.*

**We chose these organic compounds because they actually are representative of a wide range of producers. As outlined in the introduction, our aim is to explore the suitability of biomarkers from multiple sources for PCO2 reconstructions, as previously done for porphyrins and phytane. By exploring a larger groups of producers in open ocean settings, we may be able to extend the PCO2 record derived from epsilon p, as has been done for the Cretaceous (Bice et al., 2006; Sinninghe Damsté et al., 2008; Naafs et al., 2016) and for the Phanerozoic (Witkowski et al., 2018), both reconstructed from phytol's diagenetic product phytane. Although the sources of these compounds may be both terrestrial and/or marine, in an open marine setting they will almost entirely be derived from phytoplankton (and in the case of cholesterol, also the zooplankton that consume and retain the isotopic composition of these same phytoplankton). Here, we are on the coast of a small island in open ocean and have examined our GC-MS runs for the potential presence of characteristic terrestrial biomarkers to test whether the contributors of these compounds also**

include terrestrial inputs from the island. The lack of triterpenoids and long-chain alcohols typical of higher plants suggests that our source signal is overwhelmingly marine.

We added a small section (revised manuscript Page 7, lines 2-14) in the discussion (Section 4.1) to further describe and discuss the sources and why we chose these specific compounds. When these compounds are first described in the results, we make a note in the manuscript that the sources will be discussed in the discussion.

*SECOND, underestimation of reconstructed pCO2 Figure 5 and the accompanying discussion show that the reconstructed pCO2 are significantly lower than the measured values at both the Mid and High pCO2 sites by almost a factor of two. The possible reason(s) for this are not really addressed and mainly limited to "some site limitation". This issue requires a more detailed discussion particularly with regard to possible influences of different OM sources and the validity of the assumptions used for calculating the epsilon values for each compound (Section 4.2).*

In section 4.3, we reconstruct PCO2 and describe why these reconstructed values are lower than the measured high PCO2 sites, primarily focused on the novelty of using such a site and the further research required. We have discounted different OM sources due to the lack of terrestrial biomarkers.

However, we agree with the referee that several reasons could explain the underestimation and have expanded several sections to further consider alternative hypothesis. First, we expanded the text at end of 4.2 to include why epsilon f (maximum fractionation) is not fully expressed at the high CO2 site, such as species' affinity for carbon concentration mechanisms which utilize 13C-enriched bicarbonate, as well as the recent studies that show different Rubisco types may yield lower epsilon f than previously assumed (Thomas et al., 2018). Second, we expanded the discussion at the end of section 4.3 to raise the possibility of changing b value, which includes factors influencing fractionation other than $CO_2$ .

*MINOR ISSUES*

*p. 1, line 16, "general algal compounds": What does this mean? Are these compounds sources only by algae?*

As per our response in major issues 1, general algal compounds contrast to species-specific algal biomarkers, e.g. alkenones, which occur in limited number of genera. General algal biomarkers refer to compounds that are derived from a multitude of species, i.e. a large part of the phytoplankton community, presumably overwhelmingly from phytoplankton sources based on our analyses. We clarified this in the abstract and the text in the introduction on Page 3, line 10.

*p. 2, line 1, "current proxies leave much to be desired, often with large uncertainties and conflicting values": Could the authors elaborate on this, i.e. what specific issues with the current proxies do the authors have in mind and how this work would reduce these limitations?*

**We have elaborated on this on Page 2, lines 1-10. Although there has been much progress in development of PCO2 proxies, there are few proxies which can span timescales over 100 Ma. The few that can span longer periods are terrestrial biomarkers, which tend to have larger uncertainties, e.g. paleosols. PCO2 reconstructions based on epsilon p has its problems, particularly at lower PCO2 but tends to have smaller uncertainties and so if this could be applied to longer timescales, it would offer a long geological record (less influenced by local carbon cycling) and could help constrain the estimates for these older records.**

*p. 3, line 25, "SPM": What does SPM stand for?*

**We have now removed all mention of SPM, as these samples did not yield enough organic material for isotopic measurements.**

*p. 6, line 14, "lighter d13C values": a d13C value cannot be 'lighter' or 'heavier'. It is a number. Use 'lower' or 'higher' instead.*

**We changed this to higher ($^{13}$C enrichment) or lower ($^{13}$C depletion) throughout the manuscript.**

*p. 6, lines 18-19, 29-30, "the primarily diatom-limited compound loliolide": It is an very common compound derived from many sources, including macrophytic algae and terrestrial plants, so linking it specifically with diatoms is somewhat risky. Furthermore, this compound is know to be a degradation product of fucoxanthin and other carotenoids, which are also difficult to link to a particular source during paleoreconstructions.*

**We added a small section to further describe the sources of the biomarkers and, as in response to the first major comment, describe why we chose these specific compounds. We added a small section (revised manuscript Page 7, lines 2-14) in the discussion (Section 4.1) to further describe and discuss the sources and why we chose these specific compounds. When these compounds are first described in the results, we make a note in the manuscript that the sources will be discussed in the discussion.**

**Regarding the source of loliolide, it is established that it is a product primarily derived from fucoxanthin. Repeta (1988) explores possible carotenoid sources of loliolide in modern sediments and demonstrate the fucoxanthin contributes to loliolide but are unable to demonstrate a parallel**

conversion of diadinoxanthin and other carotenoid epoxides to loliolide. Fucoxanthin is found in diatoms, as well as brown seaweeds, and is not common in terrestrial plants. The vast majority of fucoxanthin in the world is derived from diatoms, which make up a vastly larger mass of producers than brown seaweeds and generally contain more than four times as much fucoxanthin as brown seaweeds. Indeed, loliolide is often abundantly found in upwelling regions where diatoms tend to dominate.

We added a sentence to further describe the different potential sources of loliolide. However, given that all the theoretical sources are based on carotenoids, these should have the same biosynthetic pathways and thus should not impact the isotopic composition of the degradation product loliolide.

*p. 9, lines17-18, "allochthonous input of sediment": Need to provide more detail here. Is it just about sediment or about are organic matter/biomarker sources with different epsilon values that would make reconstructing pCO2 more complex?*

We will add more detail here to describe what we mean by allochthonous input, here referring to the deposit of sediment that contain our organic compounds that has originated at a distance (e.g. outside of the bay) into our elevated PCO2 sites. Sediment mixed between the high PCO2 site and sediment far removed from the seep would likewise mix the epsilon p signal derived from these sediments.

*FIGURES*

*Figure1: The figure is confusing, i.e. it is difficult to know where this island is. It needs to be shown in a broader context, e.g. with a map of Japan at least. Geographic maps also typically have lines of latitude and longitude (shown as grid) along the X- and Y-axes. Also, the direction of the geographic North should be indicated.*

We have revised this map to include an insert of the larger region (i.e. Japan) with the location of the island. We have also included latitude and longitude lines on the x- and y-axis, as well as geographic North.

*Figure 2: Is it a GC-FID trace or GC-MS (TIC or SI mode, if so which m/z?)? Why not to give the names of the compounds next to the peaks rather than list them in the caption?*

This is an GC-FID trace, which is now labelled. We have put the compounds next to the peaks rather than in the caption.

*Figure 3: A), B), and C) are not shown on the plots. These need to be labelled. What are the error bars associated with the d13C values shown on the plots? Also, instead of 'Control', 'Mid', and 'High' show the actual pCO2 values.*

**We added labels for A, B, and C. The error bars are 0.5‰, as described in the text. These were difficult to see in the figure, as they all overlap with one another, but have now been added. Although we agree that actual PCO2 values would be ideal, the large fluctuations of these measured values (as pointed out in early comments) are the reason we prefer to keep the current labels.**

*Figure 4: Here and in text (p. 8, lines 18-20), explain how the errors associated with the epsilon(p) values were calculated?*

**We used the standard deviation of the many samples taken at each site, which we have now included in the text.**

[revised manuscript text omitted]